# The Water Transport System in Astrocytes–Aquaporins

**DOI:** 10.3390/cells11162564

**Published:** 2022-08-18

**Authors:** Zuoyi Zhou, Jiangshan Zhan, Qingyun Cai, Fanqing Xu, Ruichao Chai, Kalista Lam, Zuo Luan, Guoying Zhou, Sue Tsang, Markus Kipp, Wenling Han, Rong Zhang, Albert Cheung Hoi Yu

**Affiliations:** 1Key Laboratory for Neuroscience, Neuroscience Research Institute, School of Basic Medical Sciences, Ministry of Education, National Health and Family Planning Commission, Peking University Health Science Center, Beijing 100191, China; 2Department of Cardiology, Peking University First Hospital, Beijing 100034, China; 3Hai Kang Life (Beijing) Corporation Ltd., Sino-I Campus No.1, Beijing Economic-Technological Development Area, Beijing 100176, China; 4Hai Kang Life Corporation Ltd., Units 601-605, 6/F, Biotech Centre One, 9 Science Park West Avenue, Hong Kong Science Park, Shatin, New Territories, Hong Kong; 5Shenzhen International Institute for Biomedical Research, 1301 Guanguang Road, 3F Building 1B, Silver Star Hi-Tech Park, Shenzhen 518116, China; 6Institute of Anatomy, Rostock University Medical Center, 18057 Rostock, Germany; 7Department of Immunology, School of Basic Medical Sciences, NHC Key, Peking University Center for Human Disease Genomics, Peking University Health Science Center, Beijing 100191, China; 8Department of Pediatrics, Navy General Hospital, Beijing 100048, China

**Keywords:** aquaporins, astrocytes, AQP1, AQP4, water homeostasis

## Abstract

**Highlights:**

(AQPs) are transmembrane proteins responsible for fast water movement across cell membranes, including those of astrocytes.The expression and subcellular localization of AQPs in astrocytes are highly dynamic under physiological and pathological conditions.Besides their primary function in water homeostasis, AQPs participate in many ancillary functions including glutamate clearance in tripartite synapses and cell migration.

**Abstract:**

Astrocytes have distinctive morphological and functional characteristics, and are found throughout the central nervous system. Astrocytes are now known to be far more than just housekeeping cells in the brain. Their functions include contributing to the formation of the blood–brain barrier, physically and metabolically supporting and communicating with neurons, regulating the formation and functions of synapses, and maintaining water homeostasis and the microenvironment in the brain. Aquaporins (AQPs) are transmembrane proteins responsible for fast water movement across cell membranes. Various subtypes of AQPs (AQP1, AQP3, AQP4, AQP5, AQP8 and AQP9) have been reported to be expressed in astrocytes, and the expressions and subcellular localizations of AQPs in astrocytes are highly correlated with both their physiological and pathophysiological functions. This review describes and summarizes the recent advances in our understanding of astrocytes and AQPs in regard to controlling water homeostasis in the brain. Findings regarding the features of different AQP subtypes, such as their expression, subcellular localization, physiological functions, and the pathophysiological roles of astrocytes are presented, with brain edema and glioma serving as two representative AQP-associated pathological conditions. The aim is to provide a better insight into the elaborate “water distribution” system in cells, exemplified by astrocytes, under normal and pathological conditions.

## 1. Introduction

The brain consists of about 80% water [1], and its function is inextricably coupled with water homeostasis [2]. Water transport into and outside of the brain is discretely maintained by a very complex plumbing system that balances inward and outward water flux. At the macroscopic level, the brain is bathed in cerebrospinal fluid (CSF), whose main source is the secretion from the choroid plexus located at distinct places of the cerebral ventricles but is also derived from the cerebral microcirculation and metabolic water [3,4]. The absorption of CSF that circulates through the ventricles and subarachnoid space mainly occurs through the arachnoid granulations and perhaps also happens throughout the postcapillary venules and meninges and along cranial and spinal nerve root sleeves [3,5,6]. Any volume change due to water imbalance will cause more severe damage in the brain than in other organs because the skull limits the space for expansion. At the microscopic level, brain cells are bathed in the interstitial fluid passing through the blood–brain barrier (BBB). The water content in the interstitial fluid affects the regulation of cell volume and extracellular space dimensions, thus potentially increasing cell size and inducing cellular edema [1,7]. In addition, given that neuronal activity is caused by the movement of ions between intracellular and extracellular compartments, even a minor volume change would affect neuronal signal transmission. Maintaining the balance of water and ions is therefore crucial for brain functioning.

Glial cells, in particular astrocytes, which are one of the most common glial cell types in the brain [8], are essential components of the osmotic regulatory system and play crucial roles in water and ion homeostasis. Astrocytes undergo rapid volume changes during different neuronal activities, and this process can induce a redistribution of water and ions with the local swelling of astrocytes at the sites of neuronal activities together with the shrinkage at distant sites [9,10,11,12,13,14]. Astrocytes adjacent to neurons during osmotic challenges can control swelling of their end-feet to regulate vasopressin secretion in the supraoptic nucleus and the paraventricular nucleus, and to produce other osmolality-modulating hormones [15,16]. In addition, astrocyte swelling occurs not only at the cellular level but also in their subcellular organelles such as the endoplasmic reticulum, the mitochondria, as well as the nucleus. Organelle swelling has been observed in astrocytes following exposure to glutamate or ammonia under neurotoxic conditions [17,18,19,20]. However, the consequence of organelle swelling in astrocytes is yet not well characterized and warrants further studies.

Similar to most cells, astrocytes are able to internally re-adjust their cell volume in response to osmolar stress by the regulatory volume decrease/increase processes. Either or both vasogenic and cellular edema will increase tremendously the intracranial pressure in the restricted cranium space [21,22,23]. At the same time, astrocyte swelling causes the release of neurotransmitters and the reduction in extracellular space. This will lead to the accumulation of excitotoxic glutamate and thus induce life-threatening secondary damages and complications in many brain diseases such as stroke, hepatic encephalopathy, epilepsy, and traumatic brain injuries [23,24,25,26,27,28,29]. In searching for effective therapies for these diseases, a better understanding of the underlying mechanism through which astrocytes control water and ion movement in the brain is of fundamental importance.

Aquaporins (AQPs) are transmembrane protein channels. Several subtypes are highly expressed in astrocytes and their selective pores play a crucial role in water and ion homeostasis in the brain [30,31]. AQPs mediate bidirectional water flux driven by osmotic gradient across cell membranes [2,32,33], and thereby modulate water content in the CSF and blood [34,35]. Besides maintaining water balance and buffering extracellular ion concentrations, these AQPs have versatile functions in astrocytes. The expression of AQPs significantly affects cell volume regulation [36,37,38]. AQPs are also required in various activities including signal transduction, neuronal excitation [39], neurotransmission, synaptic plasticity, learning and memory [40], neurogenesis [41,42], cell adhesion and migration [43,44] and brain energy metabolism [45]. There is still a paucity of studies for a systematic and up-to-date review of AQPs in astrocytes especially their dynamic expression and subcellular localization under physiological and pathological conditions.

## 2. Expressions, Subcellular Localizations and Functions of AQP Subtypes in Astrocytes

The AQP subtypes AQP1, AQP3-9 and AQP11 have been reported to be present in different regions of the brain including the cerebral cortex, cerebellum and choroid plexus. At the cellular level, AQPs are widely expressed in different types of brain resident cells (as shown in Table 1). Although discussing the specialization of AQPs in other CNS regions such as the spinal cord is out of the scope of this review, we refer to recent papers addressing this important aspect [46,47]. Based on the unraveled pieces of evidence, expressions of AQPs are much more enriched, diverse and dynamic in astrocytes than in neurons and oligodendrocytes. Moreover, it is now known that AQPs, despite being initially reported in water transport, act as versatile performers for many other small molecules (as shown in Table 2).

Astrocytes grown in cultures adopted a more flat, less branched, and more polygonal morphology when compared to that of in vivo astrocytes [127,128]. Furthermore, the interactions between astrocytes and basement membranes can significantly affect the distribution of AQP4 towards the plasm membrane [129,130]. To better demonstrate the subcellular localization of AQP subtypes in astrocytes under different conditions, we schematically summarize the findings in Figure 1.

### 2.1. AQP1

The first discovery of the existence of a water channel protein, later called aquaporin 1 (AQP1), was made from human erythrocyte cell membrane by Benga’s group in 1985 in Cluj-Napoca, Romania, and published in 1986 [69,134]. Later, Peter Agre purified the protein of AQP1 for which he won the 2003 Nobel Prize in Chemistry [135]. AQP1 has been shown to mediate the transmembrane movement of water in several different organs, including the kidney, eye and brain [53,81,136,137]. AQP1 expression in the human brain was first observed under pathological conditions after subarachnoid hemorrhage, and in peritumoral tissue, with the protein being located mainly on the processes of reactive astrocytes [138]. AQP1 expression in the uninjured brains was absent or too low to be observed [138,139]. Nevertheless, the labs of both Ben Barres and Baljit Khakh later confirmed using transcriptome studies that mRNA of AQP1 was indeed expressed by astrocytes [140,141,142]. AQP1 transcripts and protein in cultured astrocytes were also identified, indicating localization of AQP1 on plasma membrane, in the cytoplasm compartment membrane and possibly also on the nuclear membrane [53,133]. Another interesting finding of AQP1 in culture is that the protein was selectively lost when passaging primary astrocytes, but its mRNA was always detectable [139]. The expression of AQP1 in the non-human primate brain was also found to be in the processes and perivascular end-feet of a subtype of astrocytes localized mainly in the white matter and at sites of the glia limitans [48]. So far, no isoforms of AQP1 have been reported.

Unlike other AQP subtypes, it has been suggested that AQP1 might harbor the capability of transporting cations after stimulation with adenylate cyclase activator forskolin or protein kinase A in Xenopus egg [72]. Further discoveries by the same group indicated that AQP1 is a cGMP-gated cation channel [143,144] and that it functions as both a water channel and a gated ion-channel in the choroid plexus, contributing to the regulation of CSF production [145]. Yool believed that the putative central pore in AQP1 is related to its cation transport functions [1,81,146]. However, other studies reported that such cation transport could not be repeated in similar Xenopus egg models [147,148,149,150,151,152]. In the molecular model presented by Murata and colleagues, the pore was only slightly larger than a typical water molecule, and the formation of hydrogen bonds between water and pore residues allowed only water molecules to pass through the channel [153]. Similarly, the determination of the AQP1 crystal structure enabled the elucidation of the mechanism by which water molecules moved through the protein, but the passage of protons and other ions was blocked [30]. The mechanisms for cation transport in different tissues and cell types thus still need further clarification. Nevertheless, a recent paper clearly showed that AQP1 can mediate fast swelling kinetics and play a key role in triggering and accelerating regulatory volume decrease, a process where cells adjust their cellular volume in response to swelling [154].

AQP1 was also one of the first AQP subtypes reported to facilitate the migration of endothelial cells [155] and to promote reactive astrocyte migration [139]. Cell migration is a fundamental process in multicellular organisms where tissue formation requires orchestrated movement of cells to specific locations. The migration of astrocytes is important under both, normal and pathological conditions, and the current knowledge of the mechanisms involved was summarized in our recent reviews [43,156].

### 2.2. AQP3

AQP3 was first identified in kidney medulla and colon in 1994 [85]. It was found to be enriched in kidney collecting ducts and localized on the basolateral cell membrane of their principal cells playing a role in water transport [85,157]. The existence and physiological function of AQP3 in astrocytes remains debated due to its relatively low abundance in the brain. In 2001, Yamamoto and colleagues showed that AQP3 is expressed in the brain and astrocytes at the mRNA level using reverse transcription–polymerase chain reaction [51]. At the protein level, Yang et al. reported the induction of AQP3 expression in primary astrocyte cultures with hyperosmotic exposure using Western blot [158]. Moreover, AQP3 is induced and co-localized with GFAP-positive astrocytes after permanent focal cerebral ischemia [87]. It is worth mentioning that recent studies indicate there might be a continuum existing between GFAP-expressing astrocytes and other cell types such as oligodendrocytes [159]. It warrants further studies to better elucidate the role of AQP3 in astrocytes.

### 2.3. AQP4

#### 2.3.1. Isoforms of AQP4

AQP4 was identified by Peter Agre in 1994 [88] and is currently the most studied AQP in relation to its properties in astrocytes. The expression level of AQP4 in astrocytes is the highest among all AQP subtypes [137,140,141,142]. So far, eight isoforms of AQP4 have been reported. These include AQP4a (M1), AQP4b, AQP4c (M23), AQP4d, AQP4e (AQP4 Mz), AQP4f, AQP4-△4 and AQP4ex [160,161,162,163]. Most reports on AQP4 subcellular localizations and functions refer to the two major isoforms AQP4a (M1) and AQP4c (M23), which were the earliest found to be expressed and localized in the plasma membrane, particularly at the end-feet of rat astrocytes, and have water permeabilities [164,165,166]. AQP4c is known to form higher-order structures called orthogonal arrays of particles (OAPs) in the plasma membrane, which are distinctive square arrays of particles originally seen and most densely concentrated in the plasma membrane of astrocytes [167,168,169]. AQP4a can co-assemble with AQP4c to form these OAPs albeit of smaller size [44,170,171]. Although there are as yet no reports regarding the protein of AQP4a and AQP4c in human astrocytes, studies of the human AQP4 genes in the brain did indicate two distinct mRNAs corresponding to these two functionally active isoforms found in the rat [172]. AQP4b, AQP4d, AQP4e and AQP4f were identified in the rat brain in 2008 [160]. Of these four AQP4 isoforms, AQP4b and AQP4d are localized extensively in acidic compartments such as endosomes or lysosomes, and the Golgi apparatus, but they do not contribute to OAPs in the plasma membrane in transfected rat astrocytes [173]. AQP4e was confirmed to have water permeability and is predominantly localized in the plasma membrane, with a minor amount in Golgi apparatus and degradation compartments [130,160]. Similar to AQP4a, AQP4e is unable to form OAPs by itself but can associate with AQP4c to form OAPs [161,170]. Moreover, AQP4e was found to have a novel effect on OAP reorganization and participated in the regulation of astrocyte cell volume [174]. However, due to the presence of in-frame stop codons, AQP4e was only found in rat brains and not in humans or mice [161]. AQP4f was believed to partially localize to the Golgi apparatus [160]. The water transport capacity of the isoforms AQP4b, AQP4d and AQP4f are not clear yet [137], and no information on these three isoforms in humans is available to date.

AQP4-△4 lacks exon 4 and is found in human skeletal muscle. There is no information on whether it is inherently expressed in astrocytes or not. AQP4-△4 transfected in HeLa cells resides mainly in the endoplasmic reticulum and does not show any water transport capacity [162]. Moreover, AQP4-△4 transfected into astrocytes with stable expression of functional AQP4 could reduce the expression level of full-length AQP4 in the cell membrane [162]. These findings suggest that AQP4-△4 may act as a pre-form of AQP4 in regulating water transport across cell membranes.

On the other hand, AQP4ex is characterized by a C-terminal extension that is generated by a programmed translational read-through and was recently identified in human, rat and mouse astrocytes [163,175,176]. AQP4ex appears to be similar to AQP4a and AQP4c in terms of being present in OAPs, required to address OAPs to the perivascular astrocyte end-feet, and able to modulate OAP size and water transport [163,175,176].

Since all these isoforms originated from the same pre-mRNA transcript after alternative splicing, each AQP4 isoform might play a role in influencing the total composition of AQP4 in a cell. However, it is unclear whether the proposed interactions among AQP4 isoforms would ultimately contribute to the exquisite water homeostasis function in astrocytes under both normal and pathological conditions.

#### 2.3.2. AQP4 Polarization

AQP4 polarization occurs in astrocytes, referring to its ‘polarized’ expression primarily in the perivascular astrocytic end-feet domains (Figure 1). Polarization is highly responsive to the extracellular microenvironment and has a preference for areas in contact with the fluid compartments in or around the central nervous system (CNS). This is best demonstrated in the specific occurrence of polarization in the boundary between parenchyma and major fluid compartments such as the brain–blood interface, the brain–subarachnoid CSF interface (glia limitans), and the brain–ventricular CSF interface (sub-ependymal astrocytes) [177,178]. AQP4 is diffusely expressed in astrocyte end-feet covering blood vessels throughout the hippocampus [179]. In the cerebral cortex, AQP4 is more enriched in the end-feet membranes adjacent to capillaries than in the parenchymal membranes and the degree of astrocytic polarization is species-dependent [178,180,181]. In general, the predominant subcellular localization of AQP4 is in the astrocyte end-feet membranes directly in contact with the brain capillaries and the pia mater, and with a low but significant concentration in non-end-feet membranes, such as those astrocyte membranes that ensheath glutamatergic synapses [10,178]. In the spinal cord, similar to the situation in the brain, AQP4 is primarily expressed in the astrocyte end-feet surrounding blood vessels. However, a strong AQP4 signal could also be found in astrocyte process wrapped around myelinated nerve fibers [46]. AQP4 polarization is not found in cultured cells, where AQP4 was found to be evenly distributed throughout the cell membrane under normal conditions [130,182], even though these in vitro water channels were functional [183].

#### 2.3.3. Physiological Functions of AQP4

Overall, AQP4 is considered the main channel for mediating fast fluid movement in response to osmotic changes and other special demands in the CNS [137]. AQP4-knockout reduces water permeability in primary cultures of astrocytes, and causes a 10-fold reduction in BBB water permeability in mouse brains [34,183,184,185]. This is in line with the polarized expression of AQP4 in the astrocyte end-feet membranes in the BBB and blood-spinal cord barrier [186].

AQP4 in astrocytes was also reported to play a major role in the brain-wide glymphatic system and its deletion markedly reduced CSF flux [35,187,188]. The glymphatic system was first reported and named by Maiken Nedergaard to reflect its dependence on glial cells and similarities to the peripheral lymphatic system. It is responsible for waste clearance in the CNS and consists of a paravascular pathway that facilitates CSF flow through the parenchyma to flush extracellular solutes from the interstitial compartments [189,190]. Recent studies have observed a 60% increase in extracellular space in the brain during sleep, and this was associated with increased glymphatic flow [191]. The studies on AQP4 and glymphatic clearance of harmful waste products during sleep herald a more comprehensive understanding of the pivotal roles of astrocytes in the CNS. Given that AQP subtypes other than AQP4 are also expressed in astrocytes, it would be of great interest to find out whether they have roles in the glymphatic system as well as whether these subtypes could compensate for each other in performing these functions.

AQP4 was found to concentrate in astrocyte processes around the parallel fiber synapses (i.e., the gaps between axonal extensions from cerebellar granule cells and dendritic spines of Purkinje cells), suggesting that AQP4 could participate in the functioning of tripartite synapses, formed by neuronal synapse and astrocyte processes [178]. The clearance of extracellular K^+^ by astrocytes through Kir4.1 K^+^ channels is essential for the stabilization of neuronal activity [2,192]. The knockout of α-syntrophin leads to the mislocalization of AQP4 and a delayed K^+^ clearance [193]. Moreover, AQP4-null mice showed a slower accumulation and clearance of K^+^ in the brain extracellular space during neuronal excitation [39,194,195].

Modulation of neurotransmitter uptake in tripartite synapses by AQP4 is another of its potential synapse-related functions. The main neurotransmitter implicated is glutamate. Glutamate serves as the major excitatory neurotransmitter in the vertebrate CNS [196], accounting for over 90% of the synaptic connections in the human brain, and serves as the primary neurotransmitter in neuronal–astrocytic interactions [197,198]. Glutamate uptake in astrocytes is primarily by excitatory amino acid transporters 1 and 2 (EAAT1 and EAAT2) and is an electrogenic process accompanied by the cotransport of three Na^+^, one H^+^ and over 400 water molecules, and counter-transport of one K^+^ [199,200,201,202]. In addition, glutamate activates its metabotrophic receptors to induce astrocyte membrane depolarization [203], opens the Na^+^/K^+^ channels, activates Na^+^/K^+^-ATPase, and increases intracellular Ca^2+^, Na^+^, Cl^−^ and K^+^, which results in water uptake and astrocyte swelling [18,204,205,206,207]. Glutamate-induced swelling was associated with K^+^ influx and was dependent on Na^+^ and Ca^2+^, but not Cl^−^ [208].

AQP4 was found to co-localize with EAAT2. This complex exists in astrocyte plasma membranes as a macromolecular complex that plays a major role in the clearance of glutamate from the synaptic space and uptake by astrocytes [209,210,211,212]. AQP4 also interacts with the metabotropic glutamate receptor 5 (mGluR5) and Na^+^/K^+^-ATPase in astrocyte membranes to form three-component macromolecular complexes [205], participating in the regulation of glutamate-uptake and Glu-induced astrocyte swelling partially through mGluR5 activation [213]. The AQP4 participation in glutamate uptake is likely indirect and occurs through involvement in the water-uptake process that necessarily accompanies uptake of the neurotransmitter. AQP4 has also been shown to be involved in the regulation of dopamine, hormones and other neurotransmitters [214,215] but the mechanisms involved are unclear. It also remains to be elucidated whether the other AQPs located on the astrocyte plasma membranes (e.g., AQP5 and AQP9) are involved in glutamate-mediated astrocyte swelling.

AQP4 plays a critical role in the regulatory volume decrease in astrocytes [38,154,205,216], and might interact with the transient receptor potential vanilloid isoform 4 (TRPV4) channel [217]. Immunofluorescence microscopy and immunogold electron microscopy on rat brain sections revealed enrichment of TRPV4 in astrocytic processes of superficial layers of the neocortex and their end-feet facing the pia mater and blood vessels [218]. This localization pattern is highly similar to that of AQP4, suggesting that AQP4 and TRPV4 participate as a complex in astrocytic regulatory volume decrease [38]. In addition, both cultured astrocytes from AQP4-knockout mice and astrocytes treated with TRPV4 siRNA failed to respond to hypotonic stress, supporting the need for both AQP4 and TRPV4 to induce regulatory volume decrease in astrocytes [38]. Moreover, the water influx through AQP4 was shown to drive Ca^2+^ influx via TRPV4 in the retinal Müller glial end-feet, which not only regulates the expression of AQP4 and Kir4.1 genes but also facilitates the time course and amplitude of hypotonicity-induced swelling and regulatory volume decrease [216]. Although AQP4 involvement in triggering and accelerating regulatory volume decrease was confirmed, the results of two recent reports suggest that TRPV4 may not be essential in astrocyte volume regulation [154,219]. AQP4-influenced swelling kinetics, however, are the main trigger for regulatory volume decrease and in mediating Ca^2+^ signaling in astrocytes after hypotonic stimulation [154]. This might explain the findings in another report where the interaction between TRPV4 and the actin cytoskeleton, but not TRPV4 alone, was critical for triggering regulatory volume decrease [220]. Besides TRPV4, another ion channel, the volume-regulated anion channel recently attracted more attention in astrocytic regulatory volume decrease. The homeostatic regulation of cellular volume in astrocytes by volume-regulated anion channels occurs as a consequence of AQP-mediated water influx, majorly through AQP4, into astrocytes under hypotonic conditions [221].

AQP4 also facilitates astrocyte migration [155,222]. AQP4 is evenly distributed in cultured astrocytes under normal conditions but becomes polarized to the migrating edge after the astrocytes are scratched (Figure 1). Similar polarization was observed in a stab-wound injury model in the mouse cerebral cortex [223]. In addition, astrocytes in AQP4-null mice migrated slower than in the wild-type after brain injury [224]. One mechanism proposed for AQP4-facilitated astrocyte migration involves AQP4 facilitation of rapid water influx at the leading edge of migrating astrocytes to cause local expansion of the plasma membrane and subsequent formation of cell protrusions that drive migration [155]. Alternative mechanisms proposed for AQP4-dependent cell migration include AQP-dependent changes in shape and volume [155], and interactions between AQPs and components of the cell locomotion system [225]. One potential locomotion system component is the adhesion-associated protein connexin 43, which is critical for intercellular adhesion, and the functional relationship identified between AQP4 and connexin 43 in the brain could contribute to AQP4-facilitated astrocyte migration [225,226]. In addition, AQP-dependent pinocytosis may contribute to the cell migration. In an ongoing study of our group, we observed that scratching could enhance macropinocytosis in injured astrocytes and accelerate their migrating rates, and it would be interesting to elucidate whether AQP4 has a role to play in the induction of macropinocytosis and on the other hand, whether the AQP3 and AQP7 subtypes, reported to control immature dendritic cells in concentrating macrosolutes via macropinocytosis [227], are associated with astrocyte migration.

Furthermore, AQP4 has been indicated to take part in brain development and synaptic plasticity [40,42,228]. AQP4 was revealed to be associated with adult neurogenesis, including the proliferation, fate specification, differentiation and migration of adult neural stem cells (NSCs) [42,229]. Although the mechanisms for AQP4 regulation of adult NSCs remain largely unexplored, its participation in controlling the water-exchange dynamics of NSCs in the subventricular zone was recently reported [230]. A recent finding of a much higher AQP4 expression in the cerebellum, hippocampus and diencephalon further indicated that AQP4 actions might extend to motor control and learning [54]. These studies highlight the potential roles of AQP4 in higher CNS functions but more investigations will be needed to understand the underlying mechanisms and to clarify whether the observed associations reflect direct effects, indirect effects or only accompanied effects to fully understand its actual involvement in higher CNS functions.

### 2.4. AQP5

AQP5 was first identified in rat salivary glands in 1995 [96] and has been shown to convey a high degree of membrane water permeability in corneal, pancreatic, and bronchial epithelium, the secretory cells in salivary and lacrimal glands, airway submucosal glands and type I pneumocytes of the respiratory tract [36,231,232].

AQP5 was first identified in primary cultures of rat astrocytes at the mRNA level, and its protein was observed in reactive astrocytes surrounding the lesion site of the hippocampus after middle cerebral artery occlusion (MCAO) induction in rats [51]. As expected, only low-level expression of AQP5 was observed in human normal brain tissue [233]. AQP5 protein was also detected in the piriform cortex, choroid plexus, dorsal thalamus and cerebellum Bergmann glia cells in rat brain [52]. The expression of AQP5 in primary cultures of mouse cerebral cortical astrocytes at both mRNA and protein levels was further identified by our group [53]. By using RNA-sequencing, Ben Barres’ group confirmed AQP5 expression in human astrocytes [141]. No isoforms of AQP5 have as yet been identified.

AQP5 localizes in both the plasma membrane and the cytoplasmic compartment membranes in primary astrocyte cultures [53]. The AQP5 expression was very low in 1-week primary astrocyte cultures but increased dramatically after 2 weeks [53,127,128]. Similar to AQP1 and AQP4, AQP5 is also permeable to water [7,234], and able to activate ion channel protein TPRV4 and take part in regulatory volume decrease [37]. Therefore, AQP5 in astrocytes may participate in water transportation and astrocytic regulatory volume decrease, regulating the osmotic pressure needed for neuronal functions in the brain. In addition, AQP5 can facilitate astrocyte migration. In the primary culture model we built, AQP5 increased and polarized to the migrating processes and the leading edge cytoplasmic membrane of astrocytes, and its over-expression facilitated astrocyte process elongation [53].

### 2.5. AQP8

AQP8 was initially described to be highly expressed in the testis and the liver [235] and was later reported also to be expressed in neural tissues (Table 2). Both AQP8 mRNA and protein expression have been determined in primary cultures of rat astrocytes [51,158]. The protein expression of AQP8 was observed in astrocytes in the mouse spinal cord in 2004 [103]. Later, AQP8 expression was also demonstrated in astrocytes in the rat brain [52,87]. Interestingly, considerable AQP8 mRNA was detected in adult NSCs, and its predominant expression was found in the mitochondria-enriched fraction indicating a possible role of AQP8 in regulating mitochondrial volume during NSC differentiation [59]. So far, three isoforms of AQP8 have been found: the full-length AQP8 transcript and two other splice isoforms discovered in pig testis [236]. Further investigations on AQP8 are required to clarify its expression and physiological roles in astrocytes and the NSCs.

### 2.6. AQP9

AQP9 was first identified in peripheral leukocytes in 1998 [116]. AQP9 is an aquaglyceroporin and functions in the transport of water, urea and glycerol [116,117,237,238]. It was later found to also transport mannitol, purines and pyrimidines (Table 2) [137,177,222,239,240,241,242].

AQP9 expression in astrocytes was also confirmed in transcriptome studies, and its expression was higher in rodent brains than in human brains [140,141,142]. AQP9 protein was detected in glia limitans, white matter and gray matter of the rodent and non-human primate brain [60,61] To date, two isoforms of AQP9 have been identified in the brain [239]: the long isoform, which appears to correspond to the liver isoform, is expressed throughout the astrocyte plasma membrane, whereas the short isoform is expressed in the mitochondria with higher enrichment in the inner mitochondrial membrane [60,239] and a potential role in transporting lactic acid into the mitochondria [239,243]. AQP9 is therefore very likely to be involved in brain metabolism.

In addition to AQP4 and AQP8, AQP9 was a key protein examined in NSC studies. AQP9 is expressed in adult murine subventricular zone- (SVZ-) derived NSCs [41] but it has as yet not been found in human NSCs [41,42]. It was reported that the levels of expression and cellular localization of AQP4 and AQP9 were differentially regulated upon differentiation of SVZ-derived NSCs into glial cells and neurons, indicating that AQP4 and AQP9 likely play different roles in these cells [41].

Although AQP9 expression and subcellular localization have been identified in NSCs and astrocytes, interpretation of these data should be undertaken with caution. The inconsistencies between various studies may sometimes be due to the cross-reactivity with nonspecific epitopes for some AQP9 antibodies [62,244]. Using AQP9 knockout mice, Rojek and colleagues demonstrated that AQP9 expression discovered in the mice brain could be due to artifacts related to the specificity of the anti-AQP9 antibodies used [238]. However, Mylonakou et al. showed that AQP9 mRNA was present in the cells where the same AQP9 immunoreactivity was observed through in situ hybridization [62]. In the same study, a lower level of AQP9 mRNA was found in mouse brain than in rat brain, which may well explain such discrepancy. Nevertheless, heterogeneity of AQP9 expression in different species and its existence of different splice variants need also to be considered. The development of novel tools with higher specificity would help to confirm the results of AQP9 expression and unravel the roles of AQP9 in astrocytes and CNS.

## 3. AQP_S_ and Pathological Brain Conditions

Changes of AQP levels in astrocytes have been reported in various CNS disorders [137,177,194,222,245,246,247,248,249]. There is an unmet clinical need for drug discovery targeting AQPs under pathological conditions [250,251]. The level of AQP1 has been reported to be upregulated in subarachnoid hemorrhage, contusion, Creutzfeldt–Jakob disease, ischemia, multiple sclerosis, Alzheimer’s disease, epilepsy, and brain tumors [2,133,139,177,252,253,254,255,256,257]. Expression and distribution changes of AQP4 were reported in reactive astrocyte migration (upregulated and polarized towards the leading edge of migration), brain edema (both up- and downregulation reported), brain tumor (upregulated), epilepsy (upregulated and localized towards the neutrophil) and mediating immunologic injury in neuromyelitis optica (decreased AQP4) [51,53,226,245,258,259,260,261,262,263]. In addition, the level of AQP8 was increased in brain tumors [115]. For the pathological conditions mentioned above, the roles of AQPs in brain edema and brain tumor have been most investigated.

### 3.1. AQPs and Brain Edema

#### 3.1.1. Brain Edema and Its Subtypes

Brain edema is a pressing clinical problem inevitably accompanying ischemic infarcts and intracerebral hemorrhages. Brain edema has a crucial impact on morbidity and mortality following traumatic brain injury [264]. Under brain edema, the tension created by the expansive force from swelling could induce life-threatening secondary damage and complications with mortality up to 80% [27,265,266]. Even in non-life-threatening stroke, the magnitude of brain edema is strongly predictive of the functional outcome of patients.

Brain edema is divided into cytotoxic brain edema, vasogenic brain edema, and hydrocephalic brain edema based on the underlying cellular mechanisms [1]. Cytotoxic brain edema is an increase in intracellular water content without the disruption of the BBB, and typically appears within 24–48 h of the initial ischemic or hemorrhagic insults. Vasogenic brain edema is a result of BBB damage with leakage of plasma proteins. Such conditions are most frequently seen around brain tumors [1,267,268]. Hydrocephalic brain edema is a result of the increase in the CSF pressure with BBB disruption, such as in systemic hyponatremia [269].

Although a huge amount of studies have been performed on brain edema prior to the discovery of AQPs, the new knowledge of AQPs in water monitoring allows one to view this traditional problem from a new perspective. The best-characterized AQP in astrocytes involved in brain edema is AQP4, which was recently reviewed for its expression, its roles in edema formation and water movements [270]. Here, we will present information about other AQPs in astrocytes associated with brain edema.

#### 3.1.2. AQP1 and Brain Edema

AQP1 expression in astrocytes was upregulated under various brain injuries. AQP1 expression was found to be elevated on astrocytic processes of reactive astrocytes in the brain after acute subarachnoid hemorrhage [138,271]. AQP1 expression in astrocytes was also enhanced in other ischemic brain injuries in vivo [132,133]. AQP1 upregulation and enrichment in astrocytic end-feet were observed also in astrocytes located in the edematous tissue after traumatic brain injuries in vivo and in astrocytes of primary cultures [53,139,256]. Most recently, the increase in AQP1 expression and p-p38 MAPK in astrocytes were found to be associated with brain edema. A p-p38 MAPK inhibitor effectively reduced cerebral edema and alleviated the associated AQP1 upregulation [272]. Moreover, given the important role of AQP1 in CSF production, AQP1 was proposed to participate in hydrocephalic brain edema resulting from the increase in CSF pressure and BBB disruption, and its threefold increase in expression levels in hydrocephalic brain edema supported this hypothesis [269]. Given the above evidence, AQP1 is very likely to contribute to the dynamics of brain edema formation or resolution. Furthermore, AQP1 might be a good indicator of brain edema severity and manipulation of AQP1 could be of benefit to the treatment of distinct brain edema subtypes.

#### 3.1.3. AQP5 and Brain Edema

AQP5 is also involved in ischemia-induced brain edema. AQP5 mRNA was shown to be downregulated in rat astrocytes after hypoxia treatment [51], and similar decreases in AQP5 expression but with concomitant increases in AQP4 expression were observed in an in vitro ischemia model of primary astrocyte cultures [53,273,274,275,276]. A decrease in AQP5 was also observed in the infarction areas of the cerebral cortex after MCAO injury [53]. However, in the scratch-wound model of primary astrocyte cultures and the in vivo stab-wound model to mimic traumatic injuries, AQP5 was upregulated and polarized to the leading migrating edge of astrocyte processes to facilitate astrocyte elongation as reported for AQP4 [53,223]. These studies predict that AQP5 and AQP4 might contribute differently to ischemia but similarly to trauma-induced brain edema. Nevertheless, AQP5 indeed plays a role in water homeostasis in brain edema [277].

Most AQP subtypes known to be expressed in astrocytes are thus very likely to be associated with brain edema development and/or progression even though their exact roles and the mechanisms involved remain unclear. Further research is required to clarify the different AQP roles under different injury conditions, e.g., AQP4 in different brain edema subtypes and AQP5 in traumatic versus metabolic injury, and how AQP4 and AQP5 work in relation to each other during a traumatic injury. Moreover, it is unknown whether there is an interplay between the different subtypes during ischemia and whether the other AQPs present in the brain also have roles to play in brain edema.

### 3.2. AQPs and Glioma

Glioma is a primary brain cancer that originates from glial cells or glial precursors [156]. Glioma can be grossly subdivided into astrocytoma, oligodendroglioma, ependymoma, and oligoastrocytoma [156,278,279]. Among these glioma subtypes, astrocytoma originates from astrocytes and its most malignant form is glioblastoma multiforme or WHO Grade IV astrocytoma [280], which is capable of widely infiltrating into the neighboring brain tissue, making complete tumor resection impossible and effective treatment difficult for this type of tumor [156].

#### 3.2.1. AQP1 and AQP4 in Glioma

Previous research has demonstrated an increase in AQP expression in human brain gliomas, particularly astrocytoma [281,282,283,284], suggesting AQPs may be involved in the development of glioma. Based on current evidence, we concluded three potential mechanisms. First, AQPs can promote angiogenesis in gliomas. AQP1 is overexpressed in gliomas and peritumoral tissue with its expression level positively correlating to the histological grades of astrocytoma [138,285,286]. AQP1 is also highly related to the migration and proliferation of astrocytoma [233]. AQP1-null mice caused impaired tumor angiogenesis and elevated AQP1 expression led to increased blood vessel formation in astrocytomas [283,287,288]. This may be through enhancing vascular endothelial cell migration, and the result is detrimental as it serves to “feed” the tumor and allow the tumor to grow and spread. Second, AQPs may be involved in brain tumor edema. This was indicated by the significant correlation between blood–brain barrier opening and the increase in AQP4 expression. Upregulating AQP4 expression in high-grade astrocytomas may facilitate fluid flow [284], and the correlation between AQP4 expression and the severity of tumor edema was further confirmed by magnetic resonance imaging [289]. Third, AQPs can directly promote tumor cell migration, thus enhancing the infiltration of tumor cells into the surrounding brain tissue [290]. For instance, AQP4 has been shown to facilitate both changes in tumor cell morphology via polarizing the cell lamellipodia and increasing the number and size of lamellipodia in migrating tumor cells [233,291]. More recently, the dynamics of AQP4 aggregation/disaggregation into OAPs and their link with the actin cytoskeleton were found to be determinants in glioma cell fate through altering plasma membrane dynamics to influence cell proliferation, cell migration and apoptotic potential [292].

#### 3.2.2. AQP5 in Glioma

AQP5 has been indicated to be also involved in astrocytoma development, although compared to the prominently expressed AQP1 and AQP4, its expression is low [233]. A recent report found that AQP5 expression in primary glioblastoma was associated with the tumor size and whether the excision was complete or incomplete. By gene silencing, AQP5 was shown to be essential for human glioma cell proliferation, migration and protecting the cells from apoptosis through regulating the EGFR/ERK/p38 MAPK signaling pathway [293].

#### 3.2.3. AQP8 in Glioma

AQP8 mRNA expression in human brain astrocytic tumors was found to increase in direct correlation with the pathological grade of astrocytoma. These data may suggest that AQP8 could also contribute to the proliferation and/or migration of astrocytomas [115].

Our group has previously used the scratch-wound injury model to induce astrocyte reactivation [294], and summarized the similarity of reactive-astrocyte migration in comparison with astrocytoma metastasis [156]. However, the overall mechanism of AQP-participation in tumorigenesis and the crucial roles of AQPs in brain tumor cells are still unresolved.

## 4. Conclusions

Cellular water homeostasis is a fundamental cellular function, and with the sought-after discovery of AQPs, a mechanism was provided whereby water flow in and among cells occurred in a comprehensive, well-organized manner rather than through random leakage across the cell membranes. To date, at least thirteen subtypes of AQP have been identified and various isoforms have been detected for AQP4, AQP8 and AQP9. Moreover, apart from the plasma membrane, some subtypes have been found to locate in the cytoplasm compartment membranes (e.g., AQP1 and AQP5) and in subcellular organelle membranes (e.g., AQP1 in the nuclear membrane and AQP8 in the mitochondrial membrane). The diversity of the AQP family, both in membership and in the expression and localization patterns, suggests that they are under complex regulation and that they may serve multiple functions in addition to water transport.

In the brain, astrocytes are proposed to be the principal homeostasis cells and are responsible for maintaining the balance of various small molecules such as water, ions, neurotransmitters and energy substrates. Among these actions, water homeostasis is the primary process involving astrocytic AQPs, but for the subtypes AQP1, AQP4, AQP5 and AQP9, additional roles have been indicated. In particular, extensive studies have identified a variety of astrocytic functions for AQP4 including fluid movement in the glymphatic system, K^+^ uptake, glutamate uptake and glutamate-induced swelling, the regulatory volume decrease and astrocyte migration. However, although AQPs have been identified for over two decades, they are clearly understudied. We are still at the stage of identification of their subcellular location, subtypes and isoforms, and are still far from understanding their pivotal roles under physiological (e.g., in the glymphatic system) and pathological conditions (e.g., in brain edema and brain tumor). Many subject areas still require further investigations, including but not limited to: the roles of AQPs in astrocyte volume regulation at the sub-organelle level (e.g., in the mitochondria and nucleus); the relationship between AQPs and astrocyte differentiation, injury, reactivation and gliosis in terms of cell volume, shape, migration and elasticity; the functional interaction among different AQP subtypes and their isoforms in astrocytes under both normal and pathological conditions; and the mechanisms for water transport including the regulation of direction-of-flow for water, ions and other molecules. This review serves to bring to the attention of the reader the essential existence of the AQP family and its specific composition of isoforms in regulating water homeostasis in astrocytes. Moreover, AQPs can regulate many different astrocytic functions, possibly extending to advanced CNS functions such as motor control and learning, and we wish to point out that much more effort will be required to fully understand this “water distribution system”.

## Figures and Tables

**Figure 1 cells-11-02564-f001:**
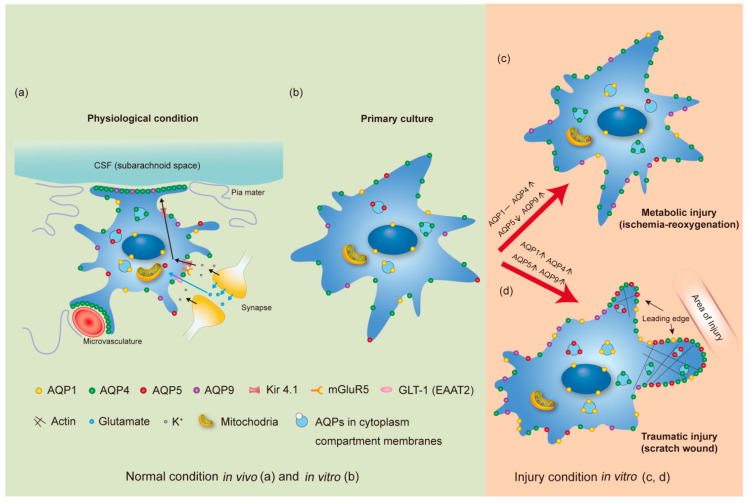
**AQP expression and subcellular distribution in astrocytes.** The schematic representation summarizes the expression levels and subcellular localization of AQP1, AQP4, AQP5 and AQP9 in astrocytes under physiological conditions in vivo (**a**), in normal primary culture condition (**b**), and in primary culture upon metabolic (ischemia-reoxygenation) injury (**c**) and traumatic (scratch-wound) injury (**d**). Different colors are used to indicate the different subtypes: AQP1 (yellow), AQP4 (green), AQP5 (red) and AQP9 (purple). The symbol density reflects the comparative expression level of AQPs in specific regions, e.g., polarization of AQP4 to the astrocyte membranes facing CSF compartments or facing the microvasculature. In part (**a**), the directions of flow of K^+^ ions (black arrows) and glutamate molecules (blue arrows) in a tripartite synapse situation are indicated. The glutamate receptor (mGluR5), the glutamate transporter (GLT1/EAAT2), and the K^+^ transporter (Kir 4.1) known to interact with AQP4 are also depicted. Changes in the expression levels of AQPs between normal in vitro and the two injury conditions are indicated by bracketed symbols as increase [↑], decrease [↓], or unchanged [-]. Although these changes are shown only for in vitro conditions, similar changes for AQP4, AQP5 and AQP9 have been reported for in vivo conditions. Divergence for AQP1 expression change was demonstrated (not shown in the figure). Stable levels of AQP1 were determined by us in primary culture of mouse astrocytes after ischemia treatment and mouse MCAO model [53,131]. However, the other two studies reported induction of AQP1 expression in astrocytes using the rat MCAO model and human ischemic brain lesions when compared to physiological conditions [132,133].

**Table 1 cells-11-02564-t001:** Expressions ^†^ of AQPs in cerebral cortex, cerebellum, choroid plexus, brain cells and neural stem cells.

AQP Isoforms	Cerebral Cortex	Cerebellum	Choroid Plexus	Astrocyte	Oligodendrocyte	Neuron	Neural Stem Cell
AQP1	Yes [48]	Yes [49]	Yes [48,50]	Yes [48]	-	Yes [48]	-
AQP3	Yes [51,52]	Yes [52]	Yes [52]	Yes [51,52]	-	Yes [51,52]	-
AQP4	Yes [53,54]	Yes [54]	-	Yes [41,53,54]	-	-	Yes [41,42]
AQP5	Yes [52,53]	Yes [52]	Yes [52,55]	Yes [53]	-	Yes [52]	-
AQP6	-	Yes [56]	-	-	-	Yes [57]	-
AQP7	-	-	Yes [58]	-	-	-	-
AQP8	Yes [51,52]	Yes [52]	Yes [52]	Yes [51,52]	Yes [51]	Yes [51,52]	Yes [59]
AQP9	Yes [60]	Yes [61]	-	Yes [60,61]	-	Yes [61,62]	Yes [41]
AQP11	Yes [63,64]	Yes [65]	Yes [64,65]	-	-	Disputed [63,64]	-

AQP expressions are summarized for the three brain areas on which most studies on AQPs in the CNS are focused. Microglia are not included as they are related to the macrophage/monocyte lineage. ^†^ Expressions of AQPs are based on detection of either mRNA or protein.

**Table 2 cells-11-02564-t002:** The tissue distribution and transport characteristics of mammalian AQPs.

AQP Subtypes	Transport Molecules	Neural Tissue Distribution
AQP0	Water [66,67,68], anions [66]	Eye (lens) [66]
AQP1	Water [69,70], gas [71], and possibly cations [72]	Peripheral nervous system [73,74,75,76], eye (cornea, retina, lens, choroid, ciliary body, iris, conjunctiva) [77,78,79,80], brain [81]
AQP2	Water [82]	Facial nerve [83], trigeminal ganglion [84]
AQP3	Water, urea, glycerol [85], ammonia [71], H_2_O_2_ [86]	Eye (conjunctiva, cornea) [79], brain [87]
AQP4	Water [88], CO_2_ [89], ammonia [90]	Brain [54,91], spinal cord [92], eye (retina, cornea, iris, ciliary body, lens) [80,93], optic nerve [94], sensory ganglion (trigeminal ganglion, dorsal ganglion) [95]
AQP5	Water [96], CO_2_ [97]	Eye (lens, retina, cornea, iris, conjunctiva) [80,98,99,100,101,102], brain [55], spinal cord [103]
AQP6	Water [97], urea [97,104], anions [105], glycerol [104], nitrate [106]	Eye (retina) [107], brain, spinal cord [108]
AQP7	Water, urea, glycerol [109], ammonia [97], arsenite [110]	Brain [111]
AQP8	Water, urea, ammonia [97,112], H_2_O_2_ [113]	Spinal cord [103], eye (lens) [114], brain [115]
AQP9	Water, urea [116], glycerol [117], lactate [118], H_2_O_2_ [119], mannitol, purines, pyrimidines [110]	Optic nerve [120], eye (retina) [121], brain [60]
AQP10	Water, urea, glycerol [122]	Undetermined
AQP11	Water [123,124]	Brain [64], eye (retina) [125,126]
AQP12	Undetermined	Undetermined

## Data Availability

Data sharing is not applicable to this article as no new data were created or analyzed in this study.

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
