# Peer review of "The Water Transport System in Astrocytes–Aquaporins"

_cells, 2022, doi:10.3390/cells11162564_

Round 1
Reviewer 1 Report
In the paper entitled “The water plumbing system in astrocytes – Aquaporins” the authors reviewed the recent advances in AQPs in astrocytes, focused in controlling water homeostasis in the brain.
The authors have put significant effort in the study of the literature about AQPs in the CNS, and have prepared tables with extensive information. However, it is mandatory that the text is proofread by an English language editor before publication.
Page 2: The text “In addition, astrocyte swelling not only happens at cellular level, but also in their subcellular organelles such as endoplasmic reticulum, mitochondria, as well as nucleus [15-18]” should be followed by a brief explanation of the consequences of organelle swelling.
Page 3: Instead of “The AQP subtypes AQP1, AQP3-9, and AQP11 have been reported to be present in the CNS. We have summarized the expressions of AQP subtypes in the cerebral cortex, cerebellum, choroid plexus, brain cells and neural stem cells (NSCs) in Table 1. Based on the unravelled evidences, expressions of AQPs are much more enriched, diverse and dynamic in astrocytes than in neurons and oligodendrocytes. In addition, we have also summarized the determined transport characteristics of AQPs in Table 2.” Please use different style of writing, and avoid the style “we have summarized…”, which is suitable for the abstract.
Page 4: “It is worth to mention that astrocyte….” – please rewrite by “Astrocytes grown in ….”
Page 5: “Furthermore, astrocytes cultured in uncoated or poly-lysine coated culture vessels have been reported to cause unrealistic subcellular localization of some membrane proteins including certain AQPs [49,50].” Please rephrase. Unrealistic?
etc. The whole text needs significant rewriting in the cohesive way.
Page 6: Why do you point out AQP3 to conclude that it is not present in astrocytes?
Please include subchapters in chapters 2.3., 3.1, 3.2 …
Reviewer 2 Report
The authors provide an ambitious and useful overview of aquaporins (AQP), focusing on their presence in astrocytes, and on the functional roles of AQP in the brain. Figure 1 is helpful, and the Tables also helps the reader to realize the potential impact of AQPs in the organism. However, the manuscript needs major revisions with regard to organization and presentation.
Major
1. Title is, to my mind, not quite adequate. “Plumbing” is sometimes used to popularize our circulatory system (blood and lymph), rather than fluid transport across membranes. I am concerned that “plumbing” in the title of this review will be a bit misleading, given the existence of the glymphatic system, which, after all, in the end relies on “tubes”.
2. The manuscript is an extensive line-up of the discovery, distribution, subcellular and molecular features, as well as functional aspects of AQPs found in the brain. The description is very compact, overwhelmed with details, and a reader who is not already very familiar with the field, will very quickly lose track of the main aspects. I recommend the authors to use subheadings for most of the different AQP sections, stating the main functional (and/or other) key points of each subheading. In this way the reader will have a chance to get an idea of what the different AQPs are doing and how their different structure, localization etc, might complement each other.
3. Although, it is of some interest to learn when, who and in which tissue each AQP was first detected, some of this information about non-neural tissue could be omitted for the sake of reader friendliness. This definitely goes for Table 2, where the CNS interested reader has to navigate through an almost endless list of other locations. Trimming this Table would also make the list of references more focused on the topic of the manuscript.
4. The role of AQP in the spinal cord needs to be updated with recent refences. This has much higher priority than comments about non-neural tissues. It should be noted that the CSF drainage of the spinal cord is not the same as for the brain.
5. Section 2.2 is confusing. First, it is stated that there is a report on AQP3 in astrocytes, but in the end is stated that there is not evidence for this. Please, clarify.
6. P. 7, 2nd par from below: “Several questions rather than answers remain attractive in the field.” Is a somewhat akward expression (minor comment). However, why the following sentences pop up at this place in the manuscript is unclar. They should belong to a section for conclusions and future directions.
7. P. 8, par 4: Explain what is the parallel fiber system. Somebody well acquainted with neuroanatomy will probably understand, but not necessarily everybody interested in AQP. Is this the only evidence that AQP may have a role in the function of tripartite synapses. This would be surprising, since they are nowadays a fundamental concept in neurophysiology. Please clarify, and, if there are additional evidence, elaborate.
8. The presentation is in many sections is poor, and expert language review is strongly recommended. Here are some examples, where I have difficulties to understand the message:
P. 9, 3rd par: “AQP4 plays a critical role in RVD in astrocytes [36,76,126,137], might as do the transient receptor potential vanilloid isoform 4 (TRPV4) channel which reacts to hypotonic stimuli with a conductance for Ca2+ [138].”
P. 9, par 4: “Compare to TRPV4, more consolidated evidence indicating that the volume-regulated anion channel (VRAC) which is a chloride channel activated upon cell swelling is critical in astrocytic RVD.”
P. 9, par 4, lower down (line numbers not available in my pdf): “Nevertheless, whether the other subunits (B-E) expressed in astrocytes and are there any association between them to AQP4 structurally or functionally still lack of further investigation.”
CONCLUSIONS, 2nd par. 2: “In the brain, astrocytes are proposed to be the principal homeostasis cells in the brain, and are responsible for maintaining the homeostasis of various small molecules such as water, ions, neurotransmitters, and energy substrates.” Repetitions: “in the brain……. in the brain”; “homeostasis ….homeostasis…”
Minor
1. There are numerous typos and also errors with common incorrect use of “the” , grammatical word forms “singular/plural” etc. Please, make a thorough review and correct.
2. P. 2, par. 2: “Glial cells, in particular astrocytes which out number any other types of brain cell,” Does it mean that all glial cells outnumber any other type of brain cell, or that astrocytes themselves outnumber any other type? Clarify and provide reference, since this numerical issue has created some controversy.
3. P. 5, par. 1, line 2: “unrealistic”, awkward and unclear what it means, please clarify.
4. P. 10, par. 2: development and synaptic plasticity are not “higher order CNS functions”, please remove this statement.
5. CONCLUSIONS: First two sentences unnecessary. Par 2: “myriad of astrocytic functions”, myriad typically refers to so many that they are difficult or impossible to count, not appropriate here, please change.
Reviewer 3 Report
Although the diagram and table are quite useful, it is obvious that this is a well worked field as far as reviews are concerned. So one has to ask whether the current paper adds much to the already well worked field. I am unconvinced that this review is a significant advance on that of Papadopoulos and Verkman Na Rev Neurosci 2012 doi:10.1038/nrn3468,
The papers of Abir-Awan, Kitchen et al Int. J. Mol. Sci. 2019, 20, 1589; doi:10.3390/ijms20071589, Trillo-Contreras et al Biomolecules 2022, 12, 530. https://doi.org/10.3390/biom12040530 Rasmussen Mestre and Nedergaard Fluid transport in Brain, Physiol Rev 102: 1025–1151, 2022 are equally comprehensive. Drug treatment of AQPs has been reviewed recently in Trends in Pharmacologic Sciences Trends in Pharmacological Sciences, January 2022, Vol. 43, No. 1 https://doi.org/10.1016/j.tips.2021.10.009. Cover similar ground
That being said, it is fairly comprehensive and brief so might be useful for advanced students.
Round 2
Reviewer 1 Report
Authors have fully addressed all comments and the revised version of the manuscript is significantly improved. I suggest that the paper is accepted in the present form.
Reviewer 2 Report
The authors have made an extensive revision, and the manuscript is in good shape.
Reviewer 3 Report
The authors have made considerable efforts to improve the English and readability. Not my major concerns but still worthwhile. The small additions on the role of AQPs in the Glymphatic system are useful and worthy of discussion. I think